# The Epidemiology of Influenza and the Associated Vaccines Development in China: A Review

**DOI:** 10.3390/vaccines10111873

**Published:** 2022-11-06

**Authors:** Jiayou Zhang, Xuanxuan Nian, Xuedan Li, Shihe Huang, Kai Duan, Xinguo Li, Xiaoming Yang

**Affiliations:** 1National Engineering Technology Research Center for Combined Vaccines, Wuhan 430207, China; 2Wuhan Institute of Biological Products Co., Ltd., Wuhan 430207, China; 3China National Biotech Group Company Ltd., Beijing 100029, China

**Keywords:** influenza epidemic, burden, vaccine

## Abstract

Influenza prevention and control has been one of the biggest challenges encountered in the public health domain. The vaccination against influenza plays a pivotal role in the prevention of influenza, particularly for the elderly and small children. According to the epidemiology of influenza in China, the nation is under a heavy burden of this disease. Therefore, as a contribution to the prevention and control of influenza in China through the provision of relevant information, the present report discusses the production and batch issuance of the influenza vaccine, analysis of the vaccination status and vaccination rate of the influenza vaccine, and the development trend of the influenza vaccine in China.

## 1. Introduction

Influenza is one of the major concerns encountered in the public health domain. It is estimated that every year, influenza leads to approximately 3–5 million cases of critical illness and 290,000–650,000 deaths related to respiratory conditions worldwide [1,2]. Seasonal influenza (influenza A and B subtypes) and highly pathogenic avian influenza (H5N1 and H7N9) greatly threaten public health and attack economies worldwide [3,4,5]. Therefore, it is essential to use the surveillance of influenza-like illnesses (ILIs) and relevant vaccination as strategies for influenza prevention and control.

All age groups are susceptible to influenza [6]. The rates of influenza vaccination are low in China, significantly lower than the rates reported for Europe and the United States [7]. The elderly and children represent the most susceptible groups due to their relatively lower immunity and heavier disease burden compared to adults [8,9,10,11,12]. Most of the influenza-related deaths (over 85%) occur among older adults aged ≥65 years. In the case of children below the age of 5 years, the world witnesses 610,000–1,237,000 cases of hospitalization due to influenza-related respiratory conditions every year [10]. While the burden of seasonal influenza manifests heavily in the cost of the medical care required, the disease also diminishes the quality of life and productivity of the affected patients. Fortunately, the risk of influenza may be mitigated by approximately 40–60% through vaccination of the general population. However, while influenza vaccination coverage is over 50% in developed nations such as Europe and the United States, it is only 2% in the Chinese population [13]. Unlike several high-income nations, China as a developing nation has not included the influenza vaccine in its national immunization program. Only in a few economically prosperous cities of China does the government provide funds for the free influenza vaccination of citizens over 60 years of age to expand the local vaccination coverage [14,15].

At present, the burden of the influenza disease and the development trend of the influenza vaccine in China are not very clear, including vaccine approval and production and the population vaccination rate. According to the relevant information on the influenza epidemic and disease burden in mainland China, by referring to the English and Chinese literature and relevant websites, and the influenza vaccine batch issuance status issued by the National Institutes for Food and Drug Control, the main objective was to describe the need for the influenza vaccine and the current developments regarding influenza vaccination and its prospects in China.

## 2. Influenza Epidemiology in China

In 1957, a national influenza center was established in China. Later, in 2000, the ILI and virological surveillance system was also established to report ILI cases and isolate the disease-causing viral strains for enhancing the knowledge repertoire of the seasonal influenza virus vaccine strains. The influenza surveillance network in China gradually expanded and improved since 2009, and now covers all prefectural and municipal hospitals, a few county hospitals, and centers for disease control and prevention (CDCs) in 31 provinces in China. The network comprises a total of 554 national influenza surveillance sentinel hospitals and 410 national influenza surveillance network laboratories. The sentinel hospitals report ILI cases to the Chinese National Influenza Surveillance Information System (CNISIS) and collect respiratory specimens. The network laboratories determine whether the collected samples are positive for the influenza virus using the real-time reverse transcription polymerase chain reaction (RT-PCR).

### 2.1. Annual and Monthly Influenza Infection and Death Cases

According to the influenza infection and death data available for mainland China for the years 2012–2021, the influenza epidemics and influenza-related deaths were concentrated mainly in the winter season (January–March and November–December) and the incidence of influenza was reported to be the lowest in the July–October period (Figure 1a). According to statistics, the numbers of seasonal influenza infections and related deaths during the high influenza season were significantly lower in the 2012–2017 period compared to those in the 2018–2020 period. The year 2019 witnessed a high influenza season with strong epidemics in the January–June and November–December periods, and the activity cycle was also extended. In addition to winter epidemics, a significant epidemic occurred in May 2020, following which the annual infection case declined. In 2021, a lower number of influenza cases were reported except for a minor epidemic in November–December (Figure 1a). It was reported that influenza-related deaths were concentrated in the January–March and November–December periods (Figure 1b), but the influenza case fatality rate showed no seasonal pattern (Figure 1c). Until 2019, the numbers of seasonal influenza infections and related deaths were observed to increase gradually with each year (Figure 1d). This could be attributed to the improved influenza CDC system, due to which the rate of reporting influenza cases increased [16]. However, after 2019, the numbers of seasonal influenza infections and related deaths became lower than the prepandemic levels of SARS-CoV-2, this could be related to the non-pharmaceutical intervention (NPI) policies of SARS-CoV-2 epidemic prevention and control, the implementation of which has also controlled the spread of influenza [17,18]. In 2021–2022, seasonal influenza vaccination campaigns were considered an important tool for reducing the proportion of susceptible people and limiting the spread of the influenza virus [19].

### 2.2. Regional and Subtype Prevalence of Influenza

Mainland China has a vast geographic area and a diverse climate and economy, due to which this region witnesses significant seasonal variation in influenza epidemics. According to the influenza surveillance reports of the 2005–2011 period, the pattern of the influenza epidemic in mainland China across the year varied due to the latitudes and geographic location of the nation. The annual periodicity of the influenza A epidemic increased with the latitude, with January–February witnessing the peak of the epidemic period in northern China (the area north of the Qinling–Huaihe line which is approximately 22° N to 34° N latitude in China), while April–June witnessed the peak of the epidemic period in regions located at <27° N. This seasonal pattern of influenza in northern China is consistent with the pattern of influenza in other temperate regions of the Northern Hemisphere in the world. In the mid-latitude provinces located within 27.4–31.3° N of the nation, influenza peaked every six months, mainly during January–February and June–August, although with a less pronounced seasonal burden [21,22,23]. The positive influenza detection results reported by the China Southern Network Laboratory for the period between 2010 and 2020 revealed that southern China (the area south of the Qinling–Huaihe line which is approximately 22° N to 34° N latitude in China) witnessed influenza epidemics in the fall and winter seasons, although a few years had positive influenza virus detections in all four seasons. In contrast, northern China witnessed a concentration of influenza detection rates in the winter and spring seasons, with positive detection rates remaining below 20% in both summer and fall seasons [24].

A systematic review and spatiotemporal analysis of the regional intra-annual seasonal variation of influenza in mainland China revealed that influenza A/H1N1pdm09 and B viruses occurred mainly during winter, while the occurrence of influenza A/H3N2 varied with the latitude, with winter epidemics observed at higher latitudes and summer epidemics at middle and lower latitudes [25]. Based on the genome analysis of H1N1, the genetic diversity of the A/H1N1pdm virus decreased from 2009 to 2017, and then increased in 2018–2019 for both influenza A (A/H1N1pdm and A/H3N2) and B strains (Victoria and Yamagata) [25]. According to the spatial and temporal dynamics analysis of influenza B virus gene sequences from 1973 to 2018 in China, the north subtropical zone (defined as 33–38° N in the Xinjiang province and 23–28° N in Hubei, Anhui, and Jiangsu provinces in China) is the origin of the B/Victoria strains, the south temperate (defined as 28–35° N in Shanxi, Hebei, and Shaanxi provinces in China) and north subtropical zones acted as transition nodes, but the southern subtropics (defined as 28–33° N in Shandong, Shanxi, and Hubei provinces) are the origins of the Yamagata lineage strains, and the northern subtropical zone and central subtropical zone (defined as 18–28° N in Sichuan, Yunnan, Guizhou, Hunan, Jiangxi, Zhejiang, and Fujian provinces in China) are the transition nodes [26]. In the period between the prepandemic and pandemic phase of SARS-CoV-2 (1 April 2019 to 4 October 2020), south China witnessed no peak summer influenza season, and the predominant subtypes/spectrum viruses in mainland China were the A/H3N2 and B/Victoria lineages. In addition, regional differences existed in the proportion of H3N2 subclades transmitted during the global epidemic of the A/H3N2 viruses [24]. The epidemiology of the seasonal influenza viruses in mainland China for the period between 5 October 2020 and 5 September 2021 in the context of the SARS-CoV-2 pandemic revealed that almost all the viruses isolated were of the B/Victoria lineage [27]. These data indicate that in different years, influenza strains were prevalent in different regions.

### 2.3. Pandemic Influenza and Highly Pathogenic Avian Influenza Systematically Reported

Prior to November 2013, H1N1 influenza was separately considered a category B infectious disease. However, after November 2013, H1N1 was transferred to category C and classified as seasonal influenza for statistical purposes, while H7N9 avian influenza was included in category B infectious disease for management purposes. The H1N1 epidemic data for the January–November periods in the years 2012 and 2013 revealed that the trends of H1N1 epidemics and seasonal epidemics were similar and concentrated in the winter–spring seasons (Figure 2a). The first case of human infection with the influenza H7N9 subtype was reported in east China in March 2013. This was followed by a certain epidemic in 2014 and 2017, with the fifth epidemic peak observed from late 2016 to early 2017. In these five epidemics, 1553 laboratory-confirmed human H7N9 cases were reported in mainland China (Figure 2b) [28]. The global Moran’s I index values for these five epidemics were 0.610, 0.132, 0.308, 0.306, and 0.336, respectively, with significant statistical differences. Spatial clusters of the H7N9 epidemic were observed in the Yangtze River Delta and Pearl River Delta regions, with a peak period from January to April. The five epidemics also expanded in scope from the east to the inner provinces and even to the west provinces of the nation [28].

## 3. Influenza Disease Burden

The burden of disease associated with influenza virus infection varies with the studied population and geographic location. Most of the available data on the burden of disease associated with influenza were from the high-income regions of the nation, while the data on the burden of influenza-associated disease in the low-income and middle-income regions were limited. According to the proportion of (influenza cases)/(infectious disease cases) and (influenza deaths)/(infectious disease deaths) from the 2012 to 2021 period (Figure 3), the proportion of influenza cases or influenza deaths to all infectious disease cases or deaths demonstrated an upward trend from 2012 to 2019 when it peaked, which was followed by a decline and a reduced proportion of (influenza deaths)/(infectious disease deaths), mainly because of the COVID-19 intervention and the improvement in influenza vaccination coverage [29,30]. The trends observed for the proportion of (influenza cases)/(infectious disease cases) and (influenza deaths)/(infectious disease deaths) over time are consistent with the chronological trends observed for the numbers of influenza cases and deaths due to influenza [29,30] (Figure 1d and Figure 3). In total, the numbers of influenza cases and related deaths are in the top five of the reported statutory infectious diseases, which shows that influenza may be a major part of the disease burden in all infectious diseases and highlights the importance of influenza vaccination.

A systematic review of the 1996–2012 data on hospitalizations due to laboratory-confirmed influenza-associated respiratory disease in children in China revealed that 8.8% of the hospitalizations due to respiratory illness were associated with influenza in children younger than 18 years, while the corresponding percentage was 7.0% in children younger than 2 years and 8.9% in children younger than 5 years. These percentages were relatively higher in northern China compared to southern China [31]. According to the three English and four Chinese databases, a review reported that the total mortality for influenza-related illness was 14.33/100,000 for all ages and 122.79/100,000 in the ≥65 years of age group, while the under 5 years of age group had the highest rates of influenza-related hospitalizations and ILI outpatient visits [32]. The Technical Guidelines for Influenza Vaccination in China (2021–2022) elaborated that the annual excess outpatient visit burden of influenza-related illnesses in China was 250 per 200,000 individuals during 2006–2015, while in 2009, during the H1N1 pandemic, the excess outpatient visit rate for ILIs reached as high as 780 per 100,000 individuals, with the heaviest burden of illness observed in children <14 years of age. It was reported that during 2010–2015, an average excess influenza-associated respiratory mortality of 88,100/100,000 occurred annually in China. The average excess respiratory mortalities due to influenza A (H1N1) pdm09, A/H3N2, and B strains were 1.6, 2.6, and 2.3 per 100,000 individuals per season, respectively. The excess respiratory mortality was estimated to be 1.5 per 100,000 individuals per season for individuals under 60 years of age and 38.5 for individuals aged 60 years or above. Approximately 71,000 influenza-associated excess respiratory deaths occurred in individuals aged 60 years or above, which accounted for 80% of such deaths [33]. Most cases of hospitalization due to severe acute respiratory infections also have other underlying conditions, such as cardiovascular disease, chronic obstructive pulmonary disease (COPD), and diabetes [34]. In COPD patients, for example, influenza vaccination reportedly reduces exacerbations, outpatient visits, hospitalizations, and mortality [35].

A systematic review of the literature from the 2006 to 2017 period reported that the numbers of excess annual influenza-related ILI outpatient visits, severe acute respiratory infection (SARI)-related hospitalizations, and respiratory deaths were 3.0 million, 2.34 million, and 90,000, respectively. The total economic burden was CNY 26.38 billion (0.266‰ of the 2019 gross domestic product (GDP)), with the hospitalization-related economic burden accounting for the highest percentage (86.4%, CNY 22.79 billion), followed by the outpatient-related economic burden. Overall, the health burden of influenza-related outpatient visits and hospitalizations was substantial. The economic burden of hospitalizations due to influenza-related SARI was higher than that of influenza-related outpatient visits and premature death, and the highest economic burden of influenza was in eastern China [36]. National sentinel surveillance data and virological data of the sentinel specimens from influenza-like illness (ILI) visits in 30 provinces of China during 2006–2015 revealed that the national average rate of influenza-related consultation was 7.8 (95% CI: 6.1, 9.6) per 1000 individuals. The burden of ILIs associated with influenza varied widely across the 30 provinces, with the highest ILI burden reported in Beijing, Tianjin, and Shanghai, while the lowest ILI burden was reported in Jilin, Ningxia, and Qinghai provinces. In children below the age of 15 years, the average influenza-associated ILI burden was 4.5 visits per 1000 individuals, which is higher than the average burden reported for those aged 15–59 years (2.3 visits) and those at the age of 60 years or above (1.1 visits). This pattern remained consistent for 10 years, with the disease burden largely attributed to influenza caused by H1N1 and H3N2. Compared to other years, the pandemic year of 2009 witnessed a higher number of influenza-related consultation rates (1000 individuals/year in China) in all provinces [37].

## 4. Influenza Vaccine Production and Batch Issuance in China

The production and use of influenza vaccines in China occurred mostly after 2000. The 2009 outbreak of the influenza A/H1N1 epidemic promoted the development of the influenza vaccine, with China leading the world in the development of the H1N1 pandemic influenza vaccine, which has quite a good level of safety and immunogenicity [38]. The influenza pandemic has driven the public to be more accepting of the influenza vaccine. Currently, most of the marketed products are seasonal influenza virus split vaccines, while only one each of the influenza subunit vaccine and live attenuated vaccine is available.

There are currently 11 manufacturers and suppliers of influenza vaccines in China, which are listed in Table 1. According to when each of these manufacturers’ influenza vaccines arrived on the market, all Chinese influenza vaccines before 2018 were trivalent influenza virus inactivated vaccines, mainly including the seasonal influenza virus split vaccine and subunit vaccine. In 2018, quadrivalent influenza virus split vaccines produced by Hualan Biological Engineering, Inc. (Xinxiang, China) arrived on the market, which was followed by the arrival of another five manufacturers with quadrivalent influenza virus split vaccines on the market in the next three years. The vaccines developed by Zhifei Longcom Biopharmaceutical Co., Ltd. (Hefei, China) (CTR20180918), Adimmune Co. (Jinan, China) (CTR20190913), Sanofi, China (Beijing, China) (CTR20191861), Dalian Aleph Biomedical Company Ltd. (Dalian, China) (CTR20200715), and Jiangsu Zhonghui Yuantong Biotechnology Co., Ltd. (Taizhou, China) (CTR20200971) have been subjected to phase III clinical trials. The influenza vaccines that are currently approved for marketing in China include the seasonal influenza virus split vaccine (IIV3), seasonal influenza virus subunit vaccine, trivalent live attenuated influenza virus vaccine (LAIV3), and quadrivalent influenza virus split vaccine (IIV4) [39].

The influenza virus split vaccine used in children aged 6 months to 3 years contains 7.5 µg of the HA antigen per dose (0.25 mL/dose) and requires dual immunization. In individuals aged 3 years or above, the influenza virus split vaccine dosage is 15 µg (0.5 mL/dose) and requires one dose. The trivalent live attenuated vaccine is used mainly for the 3–17 years age group (0.2 mL/does) in a single administration. Children’s influenza vaccines account for a minimum of 10% of the total batch share of the issued influenza vaccines [40].

In 2003, the national batch issuance system for biologics was officially implemented in China [41]. Later, in 2006, the influenza vaccine was incorporated into the biologics batch issuance management system, and the National Institutes for Food and Drug Control was responsible for publishing the number of batches of influenza vaccines issued and the number of counts per batch, but in March 2021 and later, the number of counts per batch would no longer be announced. From 2006 to 2009, the number of batches of influenza vaccines issued in China increased steadily from 1.27 million to 2.19 million [42,43]. After the launch of the quadrivalent influenza vaccine in 2018, the total number of batch issuance nationwide was 16,123,900 doses, among which the quadrivalent influenza vaccine accounted for 5,122,5000 doses. In 2019, a total of 30,784,200 doses of the influenza vaccine were issued nationwide, which included 9,710,500 doses of the quadrivalent influenza vaccine and 21,073,700 doses of the trivalent influenza vaccine. In 2020, a total of 57,519,900 doses of the influenza vaccine were issued nationwide, among which the quadrivalent influenza vaccine accounted for 33,582,300 doses, and the trivalent influenza vaccine accounted for 23,937,700 doses. The number of issued quadrivalent influenza vaccines increased every year and accounted for a relatively large proportion of all influenza vaccines issued in total (Figure 4). The trivalent subunit and trivalent attenuated influenza vaccines have only one influenza vaccine manufacturer each in China, and their batch issuance market share is relatively small.

## 5. Analysis of Influenza Vaccination Status and Vaccination Rate in China

The influenza vaccination rates are low among individuals with chronic diseases aged ≥40 years and ≥60 years in China [44]. A regular reimbursement policy for influenza vaccines, i.e., reimbursement by the local government treasury or basic social medical insurance (BSMI), could significantly increase this vaccination rate in the target population. Currently, Beijing, Shanghai, Shangdong District of Zhengzhou City, Shenzhen City, Jiaojiang District of Taizhou City, Xinxiang District of Henan Province, and Yuecheng District of Shaoxing City are offering free influenza vaccinations in China (in 2022), targeting mainly the elderly and children. Only in Shanghai has the immunization population been expanded to include teachers and health care workers as well. However, this small-scale policy approach has failed to improve overall national acceptance. A nationwide, large-scale free vaccination program requires significant annual investment and a cost–benefit analysis to determine the most effective way to increase vaccination coverage [45].

In regard to the influenza vaccine coverage among children, the vaccination rates for children aged 6–11 months and 48–59 months were lower than those for children aged 12–47 months. The top three reasons for no vaccination were the following: health care providers did not recommend the influenza vaccine, there was a lack of knowledge regarding the influenza vaccine, and people were not confident of the effectiveness of the vaccine [46]. Healthcare workers who have received the influenza vaccine before are more likely to recommend the influenza vaccination to citizens. However, questionnaire-based surveys conducted among the nurse population revealed low influenza vaccination rates [47,48]. In October 2018, the National Health Commission published instructions, according to which all hospitals were required to provide free influenza vaccinations to healthcare workers to improve vaccination rates in this population. As a consequence of the free policy and workplace influenza vaccination requirements, influenza vaccination coverage among healthcare workers was observed to increase effectively [49]. In addition, parental preferences and willingness to pay for the influenza vaccination for their children significantly influenced the rates of influenza vaccination among children. However, it was confusing that parents over 30 years of age with higher education or higher income levels were less likely to want their children to be vaccinated against influenza [50].

A meta-analysis of 126 research articles revealed that the vaccine coverage rate of the general population in mainland China was 9.4%. In addition, it was revealed that the influenza vaccination rate fluctuated widely during 2005–2017 and was much higher during 2009–2010. While the pandemic influenza vaccination rate was 37.3%, the seasonal influenza vaccination rate was 29.8% [51]. In contrast, a national cross-sectional survey conducted for the population aged 40 years and above in mainland China during 2014–2015 revealed an overall influenza vaccination rate of just 2.4% in China, which included a rate of 1.7% in the 40–59 age group and 3.8% in the 60 years and above age group [44]. When children were also included in the influenza vaccination rate results, the national average vaccination coverage was only 1.5–2.2% during 2004–2014 [45]. The main reasons for the differences in influenza vaccine coverage predictions from different researchers may be the effect of the increased influenza vaccine coverage in 2009 and differences in statistical methods.

## 6. Development Trend of Influenza Vaccines in China

The progress of China’s economy has also contributed to the development of influenza vaccines. Currently, the different types of influenza vaccines available in China include the trivalent influenza vaccine (whole-virus inactivated and split vaccine, subunit vaccine, and live nasal spray lyophilized influenza attenuated vaccine) and the quadrivalent influenza vaccine (quadrivalent influenza virus split vaccine). The current progress in vaccine development has also led to the rapid development of novel influenza vaccines, which mainly include cell-based influenza vaccines, subunit influenza vaccines, adjuvanted influenza vaccines, and live attenuated vaccines.

### 6.1. Development of Influenza Virus Split Vaccines for Different Populations

Currently, China’s domestic market mainly comprises the influenza split vaccine, while the share of the influenza subunit vaccine remains relatively smaller. The influenza vaccines available on the market are distributed mainly to the elderly population, the children, and the entire population. Hualan Biological Engineering, Inc’s quadrivalent influenza vaccine (children’s dose) has been approved and issued and is supplied to several places in China. In addition, Sinovac Biotech Co., Ltd. (Beijing, China) could soon enroll in phase I/III clinical trials for the quadrivalent influenza virus split vaccine for infants and children aged 6–35 months (CTR20220401 and CTR20220280). GDK biotechnology (Taizhou, China) has also initiated the clinical trial of the influenza vaccine for infants and children aged 3–8 years (CTR20212169 and CTR20201198). In 2019, the Wuhan Institute of Biological Products conducted a phase III clinical trial of the quadrivalent influenza virus split vaccine targeting the population aged 60 years and above (CTR20190846), and satisfactory results were obtained in terms of the vaccine’s safety and efficacy [52]. These data from the studies conducted on traditional split vaccines suggest that the development of influenza vaccines in China considers special populations, particularly the children population who have immature immune function development and the elderly populations with low immune function.

### 6.2. Novel Subunit Vaccines

In addition to the split vaccines for special populations, subunit vaccines for special populations have also witnessed certain developmental breakthroughs. The adjuvant-free quadrivalent influenza subunit vaccine, which was developed independently by Jiangsu Zhonghui Yuantong Biotechnology Co., Ltd (Taizhou, China) and is meant for individuals aged 3 years and above, has successfully completed its phase I/III clinical trials conducted in 2018–2021 (CTR20200971 and CTR20191539) [53]. Considering the safety and efficacy of this vaccine and its ability to provide broader and adequate protection against the pandemic influenza viruses during 2018–2019, the marketing application of this vaccine was accepted by the National Medical Products Administration (NMPA) of China. Yongchang Cao et al. from Jilin University linked the M2e shared sequence of the influenza A virus to the C-terminal structural domain of human serum albumin (HSA) to create a recombinant fusion protein vaccine that induced better humoral immunity in an animal model [54]. This has highlighted the feasibility of using a multi-technology platform for developing subunit influenza vaccines.

### 6.3. Development of Live Attenuated Influenza Vaccines

Live attenuated influenza vaccines are currently the only type of attenuated nasal mucosal vaccine. Since these vaccines are delivered via inhalation, the vaccines mimic the natural infection process and, therefore, provide better protection by promoting the secretion of the immunoglobulin antibody IgA from the epithelium of the upper respiratory mucosa, in addition to the normal immune response and a balanced Th1/Th2 immune response [55]. The first live trivalent lyophilized nasal spray attenuated influenza vaccine developed by Changchun BCHT Biotechnology Co. was approved for marketing in 2020, and the clinical trials revealed satisfactory results. Subsequently, in 2022, the trivalent live attenuated nasal spray influenza vaccine (non-freeze-dried) also developed by Changchun BCHT Biotechnology Co. entered phase I and phase II clinical trials (CTR20212179 and CTR20220581). The trivalent Russian-backbone live attenuated influenza vaccine was licensed in the United States in 2003, and the conversion of the trivalent live attenuated influenza vaccine to the quadrivalent live attenuated influenza vaccine was completed in 2012 [56,57]. This promoted a general confidence in the development of the quadrivalent live attenuated influenza vaccine in China.

### 6.4. Adjuvanted Influenza Vaccines

Adjuvanted influenza vaccines are currently the main direction of influenza vaccine development in China. In 1997, MF59 was approved as a trivalent influenza vaccine adjuvant in Italy [58]. Given the better efficacy and safety of MF59 in clinical trials [59], the MF59 adjuvant-containing quadrivalent influenza subunit vaccine was approved for use in the EU and the US in 2020. The results obtained using MF59 abroad stimulated several Chinese research institutions to study MF59 and other novel adjuvants. For instance, Lanjuan Li et al. used a split recombinant H7N9 influenza vaccine with a MF59 adjuvant to immunize aged mice, which induced high levels of hemagglutination inhibition and trace neutralizing antibodies and interferon-gamma responses [60,61]. The research group led by Meng Songdong used the heat shock protein gp96 as an adjuvant to immunize mice with the commercial monovalent inactivated H1N1 influenza vaccine. The team observed that this approach significantly enhanced the vaccine-specific T-cell response, effectively induced cross-reactive CD8^+^ T-cell responses against different viral strains, and activated multiple CD8^+^ T-cell responses against the conserved structural regions of the viral structural proteins NP, HA, M1, and PB1. An effective induction of protective immune responses was achieved in the mice against different subtypes of influenza viruses, such as H1N1, H3N2, and H7N9 strains [62,63]. Another study reported that the antibody titer induced with a novel emulsion (Well Adjuvant Formulation 3 (WAF3)) with the influenza vaccine was 128-fold higher than that induced in the non-adjuvant group [64]. One such emulsion is the oil-in-ionic liquid (o/IL) nanoemulsion (formulated using choline and nicotinic acid IL, squalene, and Tween 80 surfactant) developed by the research group led by Ma Guanghui, which facilitated the development of the nasal mucosal split influenza vaccine [65]. Another example is the self-assembled nanoparticles, such as aluminum nanoadjuvants, which have exhibited good adjuvant activity in several studies [66,67,68]. Another adjuvant, the lung surfactant (PS)-mimetic liposome (containing the interferon gene inducer STING agonist 2′,3′-cyclic guanosine-phosphate adenosine (cGAMP)), when formulated with the influenza vaccine, strongly enhanced influenza vaccine-induced humoral and CD8^+^ T-cell immune responses in mice and exhibited good activity against distant H1N1 and heterosubtypic H3N2, H5N1, and H7N9 viruses [69]. Although none of the adjuvanted influenza vaccines developed so far, either by the research institutes or different manufacturers, have entered clinical trials to date, it is expected that with the progress of biotechnology and further advancements in this research field, an adjuvanted influenza vaccine could soon become available in China.

### 6.5. Development of a Universal Influenza Vaccine

The Institute of Epidemiology and Microbiology in Beijing has constructed two recombinant protein vaccines, which mainly comprise a fusion peptide of the highly conserved sequences of HA and an extracellular segment of the matrix protein 2 extracellular domain (M2e) of H5N1 and H7N9, which are capable of protecting against a lethal attack from heterologous H1N1 influenza viruses [70]. This was followed by the development of a self-assembled peptide nanoparticle universal influenza vaccine with the M2e epitope linked to a fibrilizing peptide, which could protect mice against an attack from homologous H1N1 and heterologous H7N9, and owns great potential to become a universal influenza vaccine [71]. A research team from the Sun Yat-sen University prepared a recombinant H7N9 influenza vaccine by replacing the hemagglutinin transmembrane structural domain of H7N9 with the HA structural domain of H3N2, which enabled the induction of increased cross-reactive antibodies in mice [72]. The South China Agricultural University developed H7N9 virus-like particles (VLPs) containing HA, NA, and M1 proteins of the H7N9/16876 viruses and the influenza conserved epitope-based helper antigen HMN, which provided broad-spectrum antibody protection [73]. The research group led by Yaming Shan (Jilin University) expressed the influenza virus M2e and CDhelix protein in tandem with a dual-epitope self-assembled influenza nanovaccine, which induced complete protection against H1N1 attacks and partial protection against H3N2 attacks [74]. The other such vaccines developed include the intranasal nanovaccine with a conserved M2e developed by the Wuhan Institute of Virology, Chinese Academy of Sciences [75], the DNA vaccine with the genetic influenza M2e developed by the research group led by Jianjun Chen [76], and the recombinant influenza vaccine with genetic NP and M2e developed by Wenling Wang et al. [77]. All the above vaccines provide good cross-protection. These studies have indicated that the study of universal influenza vaccines in China is a multi-platform approach, which serves as a foundation for the future development of universal influenza vaccines in China.

### 6.6. Cell Culture-Based Influenza Vaccines

All influenza vaccines currently available in China are chicken embryo-based influenza vaccines [78]. However, during pandemic influenza and avian influenza outbreaks, the availability of chicken embryos was limited, which led to the emergence of mammalian cell culture-based influenza vaccine technology [79,80]. MDCK cells are suitable for influenza vaccine production in general and also in the case of certain virus strains that proliferate weakly in chicken embryos. In addition, using MDCK cells could effectively prevent the issue of allergy to chicken embryos in the population [81,82]. The cell-based production of the influenza vaccine has presented good immunogenicity [83] and could, therefore, resolve the issue of the development of various glycosylation mutation patterns during the culture of influenza virus strains in the chicken embryo [84,85,86,87]. Therefore, cell-based vaccines have become an important direction for the domestic development of influenza vaccines in China. China National Biotech Group Company Limited has conducted preclinical trials with MDCK cell-based influenza vaccines to study their immune mechanisms, while further detailed works are planned for the near future [88]. The Institute of Epidemiology and Microbiology in Beijing used an attenuated pandemic influenza virus vaccine produced in a scalable microcarrier-based MDCK cell culture bioreactor system [89] and reported achieving protection against the attacks of various lethal influenza viruses in animal models. Another strain of MDCK-derived H7N9 exhibited high growth rates when cultured in MDCK cells and a low pathogenicity when cultured in chicken embryos; subsequently, the stability and in vivo immunogenicity of the inactivated H7N9 influenza vaccine on the suspended MDCK cell matrix in animals was investigated [90]. Jiang Chunlai et al. from Jilin University detailed the process of producing the influenza H1N1 virus vaccine cultured from MDCK cells using a novel packed-bed bioreactor [91]. All the above studies provide a theoretical basis for the development of the MDCK cell matrix-based influenza vaccine [92].

## 7. Conclusions

The present report systematically analyzes the various aspects of influenza epidemics and disease burden, influenza vaccine approval and issuance, current influenza vaccination status, and future development directions of influenza vaccines in China. Over the past 20 years, China has witnessed several influenza epidemics, including the highly pathogenic avian-to-human influenza virus H5N1 in 2003, the H1N1 pandemic in 2003, and the human epidemic caused by the avian influenza virus H7N9 in 2013, which has led to a heavy disease burden on the nation. Although several manufacturers produce influenza vaccines in China, the product structure is single, and all the currently marketed products are those produced in the chicken embryo. The influenza vaccine strains produced in chicken embryos may undergo adaptive mutations, which could alter the protective effect of influenza vaccines [93]. The influenza vaccines based on MDCK cells could prevent this issue [94], although these vaccines based on cell lines are currently in their infancy stage in China. In addition, due to the decline of the immune system in the elderly, the currently administered dose of 15 µg/dose of the antigen-based influenza vaccine is not sufficient to produce adequate levels of neutralizing antibodies in the elderly [95]. Therefore, a dosage of the 60 µg/dose antigen influenza vaccine or the standard influenza vaccine with an adjuvant is recommended for the elderly population, as indicated by the research conducted with these vaccines abroad [96]. However, currently, such types of influenza vaccines are not available in China. Therefore, the development of a vaccine with a shorter production time, better protective efficacy, and broader spectrum of protection should be taken up as the current direction of influenza vaccine development in China.

## Figures and Tables

**Figure 1 vaccines-10-01873-f001:**
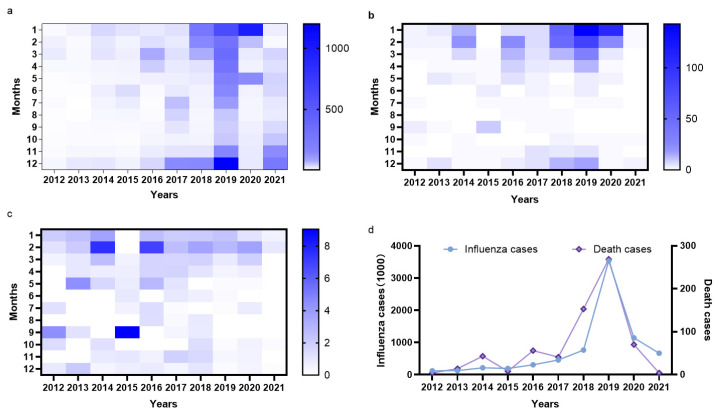
The numbers of influenza infections and related deaths in China. (**a**) Influenza cases (1000), (**b**) influenza deaths, (**c**) influenza case fatality rate (/10,000), (**d**) statistical results for annual influenza infections and related deaths (data were obtained from the Bureau of Disease Prevention and Control) [20].

**Figure 2 vaccines-10-01873-f002:**
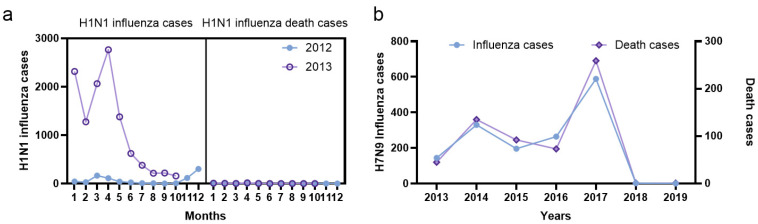
The numbers of H1N1 and H7N9 influenza infections and related deaths. (**a**) H1N1 influenza infections and related deaths in the years 2012 and 2013, (**b**) H7N9 influenza infections and related deaths during the 2013–2019 period (data were obtained from the Bureau of Disease Prevention and Control) [20].

**Figure 3 vaccines-10-01873-f003:**
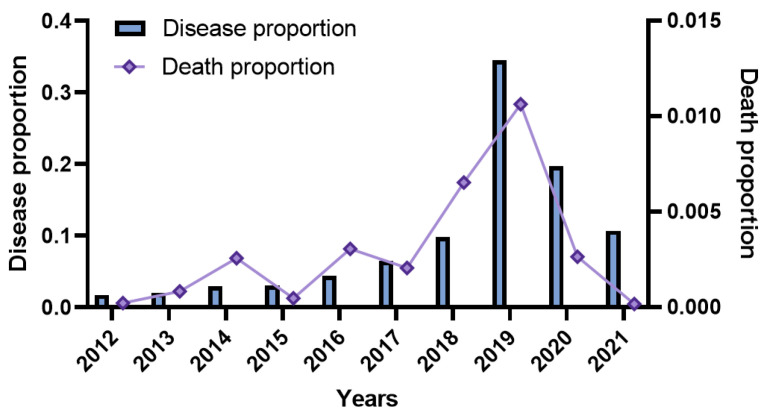
Proportion of influenza-like cases or related deaths to all infectious disease cases or deaths (data were obtained from the Bureau of Disease Prevention and Control) [20].

**Figure 4 vaccines-10-01873-f004:**
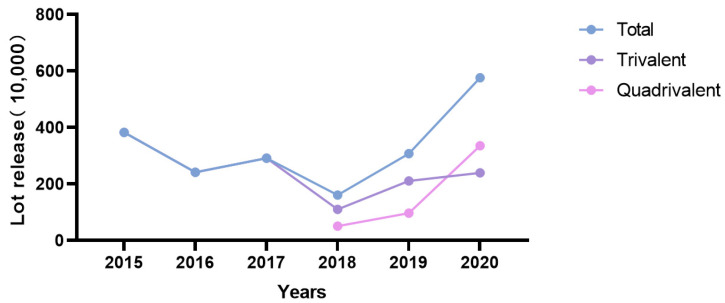
Batches of influenza vaccines issued in different years.

**Table 1 vaccines-10-01873-t001:** The list of influenza vaccine issuance manufacturers for the year 2021.

Manufacturer	Type of Vaccine	Inoculation Method	Specification (mL)	User Group	Dose of Vaccination	Time to Market
Fosun Pharmaceutical	Trivalent split	Intramuscular	0.25/0.5	6M-3Y/>3Y	2/1	2005
Sanofi China	Trivalent split	Intramuscular	0.25/0.5	6M-3Y/>3Y	2/1	2013
Adimmune Corporation	Trivalent split	Intramuscular	0.25/0.5	6M-3Y/>3Y	2/1	2015
Zhongyianke Biotech. Co., Ltd.	Trivalent subunit	Intramuscular	0.5	3Y	1	2010
Changchun BCHT Biotechnology Co.	Trivalent attenuated (nasal spray)	Spray	0.2	3–17Y	1	2020
Hualan Biological Engineering, Inc.	Trivalent split Tetravalent split	Intramuscular	0.25/0.5 0.5	6M-3Y/>3Y >3Y	2/1 1	2011/2008 2018
Sinovac Biotech Ltd.	Trivalent split Tetravalent split	Intramuscular	0.25/0.5 0.5	6M-3Y/>3Y >3Y	2/1 1	2005 2020
Changchun Institute of Biological Products Co., Ltd.	Trivalent split Tetravalent split	Intramuscular	0.25/0.5 0.5	6M-3Y/>3Y >3Y	2/1 1	2007/2004 2020
Shanghai Institute of Biological Products Co., Ltd.	Trivalent split Tetravalent split	Intramuscular	0.25/0.5 0.5	6M-3Y/>3Y >3Y	2/1 1	2004/2001 2021
GDK Biotechnology Co., Ltd.	Tetravalent split	Intramuscular	0.5	>3Y	1	2019
Wuhan Institute of Biological Products Co., Ltd.	Tetravalent split	Intramuscular	0.5	>3Y	1	2020

## Data Availability

Not applicable.

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
