# Peer review of "The Epidemiology of Influenza and the Associated Vaccines Development in China: A Review"

_vaccines, 2022, doi:10.3390/vaccines10111873_

Round 1

Reviewer 1 Report

This is an interesting and timely review that demonstrates in a condensed form how the Chinese leadership cares about the health of the population and how the Chinese healthcare system is developing. However, there are some points which should be revised.

Point 1. Line 21-23, 28-29. Please, please support this statement with a reference.

Point 2. line 367, line 370 and Table 1. Please, replace “Bcht” with “BCHT”

Point 3. Figures 2 and 3 are not directly related to the topic of the article. If the authors added another curve denoting morbidity and mortality in the vaccinated, it would be logical and more interesting.

Point 4. Figure 1. Please clarify in figure legend that this figure is for China.

Point 5. Is there any difference in morbidity and mortality in urban and rural China?

Point 6. Line 46-50. I agree with the statement “The main objective was to describe the current developments regarding influenza vaccination and its prospects in China.” However, the previous sentence “The present review aimed to collect relevant information on the influenza epidemic and disease burden in mainland China by referring to English and Chinese literature and relevant websites, the influenza vaccine batch issuance status issued by the National Institutes for Food and Drug Control” has nothing to do with the purpose of the article presented in lines 49-50, nor with the article title.

Point 7. Paragraph 1 (Influenza epidemiology in China) and 2 (Influenza disease burden) are four and half pages but does not fit to the title and main goal of the article. If the authors intend to keep these sections in the text, they should reconsider the title of the article and its main aim.

Point 8. Line 349. At the authors sure that “Ab&B BIO-TECH CO.,LTD.JS” is written correctly?

Point 9. Line 371. The authors better replace “marketed” with “licensed.” It will be more correct.

Point 10. 371-374. The statement that the American live vaccine "...promotes general confidence in the development of the quadrivalent live attenuated influenza vaccines in China" is not entirely true, since BCHT uses vaccine strains for live attenuated influenza vaccine prepared not by American, but by Russian technology. Please, correct.

Point 11. Conclusion is too long. Actually, the last paragraph (Line 495-504) is already a good and condensed conclusion. The rest of the text should be moved into the relevant sections of the review.

Author Response

eview1

This is an interesting and timely review that demonstrates in a condensed form how the Chinese leadership cares about the health of the population and how the Chinese healthcare system is developing. However, there are some points which should be revised.

Point 1. Line 21-23, 28-29. Please, please support this statement with a reference.

Answer: Thanks for the advice, the reference has been added.(Line 21-23, 28-29)(new reference1-5)(Line 28-29)(new reference6)

Point 2. line 367, line 370 and Table 1. Please, replace “Bcht” with “BCHT”

Answer: This has been revised.

Point 3. Figures 2 and 3 are not directly related to the topic of the article. If the authors added another curve denoting morbidity and mortality in the vaccinated, it would be logical and more interesting

Answer:Thanks for the advice, the title of the paper has revised as “The Epidemiology of Influenza and the Associated Vaccines Development in China: A Review, and the topic of the section 1 is more related to the topic. As the data of the morbidity and mortality in the vaccinated is limited in China, so no related statement is provided.

Point 4. Figure 1. Please clarify in figure legend that this figure is for China.

Answer: This has been revised.

Point 5. Is there any difference in morbidity and mortality in urban and rural China?

Answer: There no data for the influenza morbidity and mortality in urban and rural China, the main reason may because of the limited surveillance data and the nation disease analysis system especially in rural.

Point 6. Line 46-50. I agree with the statement “The main objective was to describe the current developments regarding influenza vaccination and its prospects in China.” However, the previous sentence “The present review aimed to collect relevant information on the influenza epidemic and disease burden in mainland China by referring to English and Chinese literature and relevant websites, the influenza vaccine batch issuance status issued by the National Institutes for Food and Drug Control” has nothing to do with the purpose of the article presented in lines 49-50, nor with the article title.

Answer: This sentence has been revised (lines 45-47), and for the line 49-50, collect relevant information on the influenza epidemic and disease burden in mainland China by referring to English and Chinese literature and relevant websites, the influenza vaccine batch issuance status issued by the National Institutes for Food and Drug Control” was not the main purpose but the important methods, and we has revised it.

Point 7. Paragraph 1 (Influenza epidemiology in China) and 2 (Influenza disease burden) are four and half pages but does not fit to the title and main goal of the article. If the authors intend to keep these sections in the text, they should reconsider the title of the article and its main aim

Answer:Thanks for the advice, the title has been revised, and the Paragraph 1 (Influenza epidemiology in China) and 2 (Influenza disease burden) fit the title and main goal of the title, which is the driving force behind influenza vaccine development, so we decide to remain this part.

Point 8. Line 349. At the authors sure that “Ab&B BIO-TECH CO.,LTD.JS” is written correctly?

Answer: This has been revised (line 269)

Point 9. Line 371. The authors better replace “marketed” with “licensed.” It will be more correct

Answer: This has been revised(line 392)

Point 10. 371-374. The statement that the American live vaccine "...promotes general confidence in the development of the quadrivalent live attenuated influenza vaccines in China" is not entirely true, since BCHT uses vaccine strains for live attenuated influenza vaccine prepared not by American, but by Russian technology. Please, correct

Answer:This sentence has been revised (line 392).

Point 11. Conclusion is too long. Actually, the last paragraph (Line 495-504) is already a good and condensed conclusion. The rest of the text should be moved into the relevant sections of the review.

Answer: Thanks for the advice, The conclusion has been regulated.

Reviewer 2 Report

Known in the field based on previous literatures:

 1. Influenza is an infection of the respiratory system mainly include nose, throat, and lungs. It is commonly called flu.

2. The vaccine comes in inactive and weakened viral forms and depending on the type they can be injected into a muscle, sprayed into the nose, or injected into the middle layer of the skin.

In this review authors cited following findings:

I have gone through the review titled “The Production and Application of Influenza Vaccines in China: A Review”. Authors nicely mentioned many information and the current developments regarding influenza vaccination and its prospects in China. The core points mentioned by authors are-

1.    As per the epidemiology of influenza in China, the nation is under a heavy burden and authors revealed many information and systemic description of the disease condition of influenza in China. 

2.  Authors cited and discussed developmental trend of the influenza vaccine, including its approval, production, and administration to the population.  

The facts and material presented are interesting and generally supportive of the conclusions drawn. There are, however, some issues that require the authors' attention. The following suggestions if incorporated could help in the better understanding of the significance of the work and implications.

Minor Concerns:

1. Figure 3, in early year (2014-2018) the morbidity proportion was high as compared to incidence but later (2019-2021) it was lower down. Are they vaccinated any time before? Explain and discuss the reason.

2. Does this review embrace a specific gap in the field as compared to previously reported review? Please explain how this review different from rest.

3. I have observed some typing errors (full stop- line 354 and spacing- line 331). Please correct them and make sure all are in one format.

Author Response

Minor Concerns:

  1. Figure 3, in early year (2014-2018) the morbidity proportion was high as compared to incidence but later (2019-2021) it was lower down. Are they vaccinated any time before? Explain and discuss the reason.

Answer: The ordinate of the incidence proportion(left) and morbidity proportion (right)is different,so all the morbidity proportion was much lower when compared to incidence, Now we has revised the incidence proportion with disease proportion, and the morbidity proportion with death proportion. As for the trends in the number of cases and deaths differ, according to the Fig1d, after 2019, the number of seasonal influenza infections and related deaths became lower than the pre-pandemic levels of SARS-CoV-2, this could be related to the nonpharmaceutical interventions (NPIs) policies of SARS-CoV-2 epidemic prevention and control, the implementation of which has also controlled the spread of influenza(line 83-87), and the reduced death proportion mainly because of the COVID-19 intervention and the improvement influenza vaccination coverage and the aroused attention to the related symptoms of the influenza disease(line166-168),

  1. Does this review embrace a specific gap in the field as compared to previously reported review? Please explain how this review different from rest

Answer: The incidence of influenza in China was different among different studies, and in our paper, we explained the difference (line323-335).,There is few such review for the vaccine produce production specific to a country (China), though some influenza epidemic has been announced in China, there are no reports that combine the predicted prevalence with the prevalence of influenza in the surveillance network.,There is no such systematic and detailed article combine the Influenza epidemic and development of influenza vaccine in China.

  1. I have observed some typing errors (full stop- line 354 and spacing- line 331). Please correct them and make sure all are in one format(?).

Answer:Thanks for the advice,this has been unified.

Reviewer 3 Report

The intention to make a summary report describing the epidemiology of influenza and the development and application of influenza vaccines in China is laudable. However, the scope seems too broad for a journal article. A monograph would be more appropriate. The section on influenza epidemiology is very weak. The statistics are poorly described, contain obvious errors, and the referenced source of the data (China's Bureau of Disease Prevention and Control) is not a published journal article or document. I believe that the authors present their own analysis of raw data from the Bureau. If so, this material needs to be put into a separate article, with clear description of the objectives and statistical methods, and presentation of summary statistics in tables and charts with sufficient detail so that readers can understand and verify the results and conclusions. As for the present manuscript, it would be much improved if the authors narrow the scope by deleting the defective parts: (L51-92 and Figure 1; and L157-172 and Figure 3) about influenza epidemiology. The remainder of the manuscript might then be acceptable.

The article in its present form is not suitable for publication. 

Influenza epidemiology in China

P1, L64: Replace "trends" with "incidence". I don't see any trend analysis of incidence rates. You are simply describing patterns.

P2, L65: Replace "prevalence" with "incidence".

P2, L73: You say there were 140,000 cases of influenza in 2019. But Figure 1a shows in 2019 there were fewer than 1,200 cases per month, therefore less than 14,400 during the year. Figure 1d shows there were some 3.8 million cases in 2019. Where do these numbers come from? Were they calculated for the same time periods and geographic regions? What is the denominator for the case fatality scale on the right of Figure 1c? Reference 9 is not verifiable. It is not possible for data obtained in 2018 to pertain to the time period 2012 to 2021. Section 1.1 and Figure 1 simply do not make sense. I recommend deleting L51-92 and Figure 1.

P3, L101: Replace "epidemiological trend" with "seasonal pattern".

P3, L101-102: Replace "the northern hemisphere of  China" with "northern China".

P3, L102: Replace "epidemiological trends" with "pattern".

P3, L105-106: Delete the meaningless clause, "and is more reflective of the tropical and subtropical regions".

P3, L108: Replace "southern hemisphere" with "southern region", and state your definition (latitude range) of the southern region of China.

P3, L110: Replace "Northern Hemisphere" with "northern region", and state your definition (latitude range) of northern region of China.

P3, L121: State your definition (latitude range) of the "North subtropical" zone of China.

P3, L122: State your definition (latitude range) of the "South temperate" zone of China.

P4, L123: State your definition (latitude range) of the "southern subtropics" of China.

P4, L124: State your definition (latitude range) of the "central subtropical" zone of China.

P4, L157: Replace "mortality" with "disease".

P5:  Figure 3 does not make sense. What are the definitions of "Incidence proportion" and "Morbidity proportion"? Do you mean [influenza cases + deaths]/[infectious disease cases + deaths], or do you mean [influenza cases]/[infectious disease cases]"? Does one of these terms pertain to mortality. i.e., [influenza deaths]/[infectious disease deaths]? In L166-168 of the text, you seem to indicate that one of the curves in Figure 3 represents absolute numbers of incident cases, not a proportion. What are the units for the scales to the left and right of Figure 3? This is all very confusing. I recommend deleting L157-172 and Figure 3.

Author Response

Influenza epidemiology in China

P1, L64: Replace "trends" with "incidence". I don't see any trend analysis of incidence rates. You are simply describing patterns.

Answer: Thanks for the advice, this has been revised (line##).incidence rates, all the data is collected from the influenza surveillance network in China(covers all prefectural and municipal hospitals and a few county hospitals, and Centers for Disease Control and Prevention (CDCs) in 31 provinces in China), it is difficult to for the surveillance network to reflected all regions, nor could it identify all infected persons. Its role is to provide an infection reference, and based on the surveillance network, we can know the general epidemic situation of influenza in China, so the article does not show the incidence rate based on the surveillance network,but with the software calculation, we emphasize that some articles report the epidemic trend,

P2, L65: Replace "prevalence" with "incidence".

Answer:prevalence has been revised with infection and death case (line 67).

P2, L73: You say there were 140,000 cases of influenza in 2019. But Figure 1a shows in 2019 there were fewer than 1,200 cases per month, therefore less than 14,400 during the year. Figure 1d shows there were some 3.8 million cases in 2019. Where do these numbers come from? Were they calculated for the same time periods and geographic regions? What is the denominator for the case fatality scale on the right of Figure 1c? Reference 9 is not verifiable. It is not possible for data obtained in 2018 to pertain to the time period 2012 to 2021. Section 1.1 and Figure 1 simply do not make sense. I recommend deleting L51-92 and Figure 1.

Answer: I am sorry for the unclear expression, for the Fig to be better to see, the ordinate data has been divided by 1000 in Fig 1a, and multiplied by 10000 in Fig 1c, all the data was analyzed based on the collected data from Bureau of Disease Prevention and Control,and the reference 9 has been revised.( http://www.nhc.gov.cn/jkj/pgzdt/new_list_12.shtml), After entering the web page, in the topic Dynamic work, we could found the overview of notifiable infectious diseases in different months and years, which including the influenza information, and all the data analyzed is based on this.

P3, L101: Replace "epidemiological trend" with "seasonal pattern".

Answer: This has been revised (line106)

P3, L101-102: Replace "the northern hemisphere of China" with "northern China"

Answer: This has been revised (line107)

P3, L102: Replace "epidemiological trends" with "pattern".

Answer: This has been revised (line107)

P3, L105-106: Delete the meaningless clause, "and is more reflective of the tropical and subtropical regions".

Answer: This has been deleted (line110)

P3, L108: Replace "southern hemisphere" with "southern region", and state your definition (latitude range) of the southern region of China.

Answer: This has been revised with southern China (line 115).

 P3, L110: Replace "Northern Hemisphere" with "northern region", and state your definition (latitude range) of northern region of China

Answer: This has been added(line115).

P3, L121: State your definition (latitude range) of the "North subtropical" zone of China.

Answer: This has been added (line127).

P3, L122: State your definition (latitude range) of the "South temperate" zone of China

Answer: This has been added (line129).

P4, L123: State your definition (latitude range) of the "southern subtropics" of China

Answer: This has been added (line132).

.

P4, L124: State your definition (latitude range) of the "central subtropical" zone of China.

Answer:This has been added (line133)

P4, L157: Replace "mortality" with "disease".

 Answer:This has changed.(line 163)

P5:  Figure 3 does not make sense. What are the definitions of "Incidence proportion" and "Morbidity proportion"? Do you mean [influenza cases + deaths]/[infectious disease cases + deaths], or do you mean [influenza cases]/[infectious disease cases]"? Does one of these terms pertain to mortality. i.e., [influenza deaths]/[infectious disease deaths]? In L166-168 of the text, you seem to indicate that one of the curves in Figure 3 represents absolute numbers of incident cases, not a proportion. What are the units for the scales to the left and right of Figure 3? This is all very confusing. I recommend deleting L157-172 and Figure 3.

Answer: I am sorry that the statement is not clear enough, this part has been revised, [influenza cases]/[infectious disease cases], [influenza death]/[infectious disease death].which means in the total infectious disease case, what is the proportion of the influenza case, in the total infectious disease death, what is the proportion of the influenza death,which shows that influenza may be a major part of the disease burden in all infectious diseases and the importance of influenza vaccination. There no units for the proportion

Round 2

Reviewer 3 Report

The manuscript has improved, but still needs some revisions before publication.

Influenza epidemiology in China

P2, L74-75: Delete the clause "which reported over 140,000 cases of seasonal influenza" because the number does not agree with Figures 1a and 1d, and it is not supported by any referenced source.

 P2, L77: Replace "the epidemic trend stabilized" with "annual incidence declined."

 P2, L80-81: Replace "although no monthly trend of influenza case fatality rate per year" with "but the influenza case fatality rate shows no seasonal pattern."

Author Response

Comments and Suggestions for Authors

The manuscript has improved, but still needs some revisions before publication.

Influenza epidemiology in China

P2, L74-75: Delete the clause "which reported over 140,000 cases of seasonal influenza" because the number does not agree with Figures 1a and 1d, and it is not supported by any referenced source.

Answer: Thanks for the advace, this has been delected(line 76)

 P2, L77: Replace "the epidemic trend stabilized" with "annual incidence declined."

Answer:the epidemic trend stabilized has been replaced with "annual infection case declined”.

This Fig1a provided the infection case, not the incidence. (line 77)

 P2, L80-81: Replace "although no monthly trend of influenza case fatality rate per year" with "but the influenza case fatality rate shows no seasonal pattern."

Answer:Thanks for the advice, this has been replaced.(line 81)

Round 3

Reviewer 3 Report

My previous comments were addressed. I do not have any other comments.